# In situ atomic-scale observation of grain size and twin thickness effect limit in twin-structural nanocrystalline platinum

Lihua Wang[1,5], Kui Du [2,5], Chengpeng Yang[1], Jiao Teng[3], Libo Fu[1], Yizhong Guo[1], Ze Zhang[4✉] & Xiaodong Han[1✉]

Twin-thickness-controlled plastic deformation mechanisms are well understood for submicron-sized twin-structural polycrystalline metals. However, for twin-structural nano-crystalline metals where both the grain size and twin thickness reach the nanometre scale, how these metals accommodate plastic deformation remains unclear. Here, we report an integrated grain size and twin thickness effect on the deformation mode of twin-structural nanocrystalline platinum. Above a ~10 nm grain size, there is a critical value of twin thickness at which the full dislocation intersecting with the twin plane switches to a deformation mode that results in a partial dislocation parallel to the twin planes. This critical twin thickness value varies from ~6 to 10 nm and is grain size-dependent. For grain sizes between ~10 to 6 nm, only partial dislocation parallel to twin planes is observed. When the grain size falls below 6 nm, the plasticity switches to grain boundary-mediated plasticity, in contrast with previous studies, suggesting that the plasticity in twin-structural nanocrystalline metals is governed by partial dislocation activities.

[1] Beijing Key Lab of Microstructure and Properties of Advanced Materials, Beijing University of Technology, 100022 Beijing, China. [2] Shenyang National Laboratory for Materials Science, Institute of Metal Research, Chinese Academy of Sciences, 110016 Shenyang, China. [3] Department of Material Physics and Chemistry, University of Science and Technology Beijing, 100083 Beijing, China. [4] Department of Materials Science, Zhejiang University, 310008 Hangzhou, China. [5] These authors contributed equally: Lihua Wang, Kui Du. ✉email: zezhang@zju.edu.cn; xdhan@bjut.edu.cn

The mechanical properties (such as strength, ductility, hardness, etc.) of a material are directly related to their atomic-scale deformation mechanism[1–5]. Revealing the atomic-scale deformation mechanism of materials is important for understanding their mechanical performance and realising their desired mechanical properties. In recent years, the atomic-scale deformation mechanism of twin-structural metals has attracted considerable interest because twin-structural metals always exhibit excellent mechanical properties[6–13]. The deformation mechanisms of twin-structural coarse-grained[1,2] and sub-micrometre sized polycrystalline metals[1,2,4,5,7–14] have been extensively studied by transmission electron microscopy (TEM), which has revealed that the twin boundaries (TBs) are not only more stable against sliding or diffusion than conventional grain boundaries (GBs) but can also obstruct the partial dislocation that glides on the plane that is inclined to the TBs (strengthening deformation model), thus making twin-structural metals very successful in terms of strength. For twin-structural nanocrystal-line (NC) metals, in which both the grain size and twin thickness (TT) are decreased to the nanometre scale, their atomistic deformation mechanisms remain unclear because direct atomic-scale observations are rarely acquired in experiments. Currently, our understanding of the atomic-scaled deformation mechanism of twin-structured NC metals normally relies on molecular dynamic (MD) simulations[15–18]. A classic MD simulation reported that the twin thickness (TT) can significantly affect the deformation model of face-centred cubic (FCC) metals[16]. For a given grain size, there exists a critical TT; for a TT above this point, partial dislocations nucleate and form TB-GB intersections and glide on those {111} planes that are inclined to the {111} TB, leading to strengthening. For a TT below this point, partial dislocations glide on the {111} planes that are parallel to the TBs, leading to softening. This mechanism has been supported by many other classic MD simulations[16,18–25] and thus promotes the widespread belief that the strengthening/softening of twin-structural NC metals is governed by partial dislocation behaviours, full dislocations and GB-mediated plasticity (softening deformation model), even though they are rarely observed in experiments. Indeed, these MD simulations can provide important information for understanding the deformation mechanisms of twin-structural metals. However, due to the lack of atomic-scale direct experimental evidence[26–29], it remains uncertain

whether these results are valid in experimental conditions. Thus, atomic-scale dynamic experiments of twin-structured NC metals are important for fully understanding their deformation mechanisms.

In this study, using a homemade double-tilt high-resolution (HR) TEM tensile stage[30,31], the plastic deformation mechanism of twin-structural NC Pt thin films are investigated in situ at the atomic scale. We discover that both the grain size and TT can significantly impact the deformation model, and some of the deformation mechanisms a not observed in twin-structural NC metals.

## Results

**Microstructure of the Pt NC thin film.** The atomic-scale deformation experiments were conducted with a homemade tensile stage (as schematically illustrated in Supplementary Fig. 1a–c). The TEM investigations show many nanosized and equiaxed grains without obvious preferred orientation (Supplementary Fig. 1d shows a bright-field image). Most grains with a $d$ ranging between 4 and 30 nm are separated by high-angle GBs, and many grains contain growth twins (as indicated by arrows) with a TT ranging from ~1 to 15 nm. Our extensive HRTEM investigations indicated that before the thin film experiences tensile strain, the TBs are atomically flat, and no pre-existing dislocations were observed (refer to Supplementary Fig. 1e–h).

**In situ observation of deformation mechanisms in twin-structural nanograins.** Figure 1 shows two HRTEM images captured during tensile loading of a $10 \times 16$ nm nanograin, which shows the full dislocation nucleation process in a relatively thick twin/matrix. Figure 1a shows an HRTEM image captured before the elastic limit is reached, in which the grain is free of defects. Figure 1b shows an HRTEM image taken 10 s after taking Fig. 1a during loading, in which two full dislocations (marked as T) gliding on the {111} planes intersecting with the TBs are observed in a relatively thick twin/matrix (originally defect-free). As noted by T, there is one set of {111} planes with an extra half plane inserted, indicating that full dislocations were blocked by the TB. In fact, our extensive HRTEM investigations show that, in relatively thick twins/matrixes, full dislocation is commonly observed, and most of them glide on the {111} planes intersecting

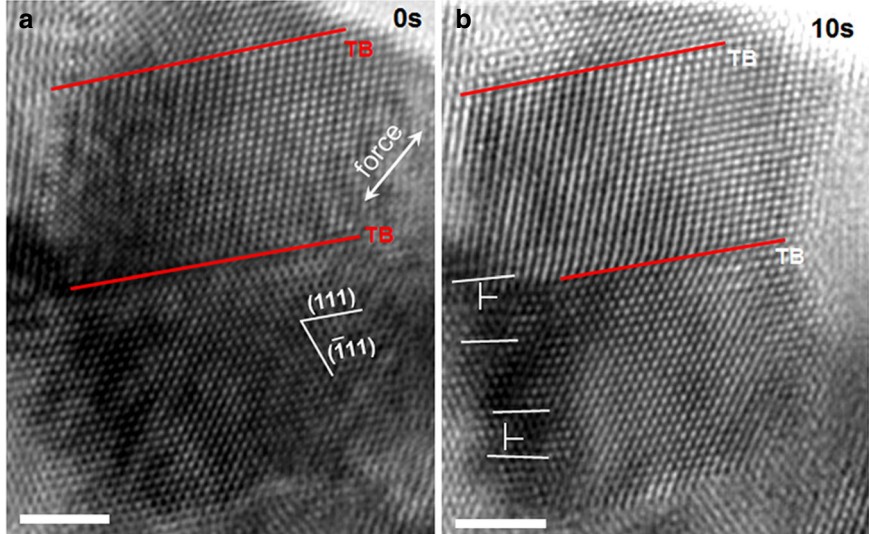

**Fig. 1 In situ observation of full dislocation nucleation in a grain that contains both relatively thick and thin twins/matrixes. a** The relatively thick twins/matrixes are free of defects. **b** Two full dislocations (marked as T) glide on the {111} planes intersecting with the TBs. The scale bars are for 1 nm.

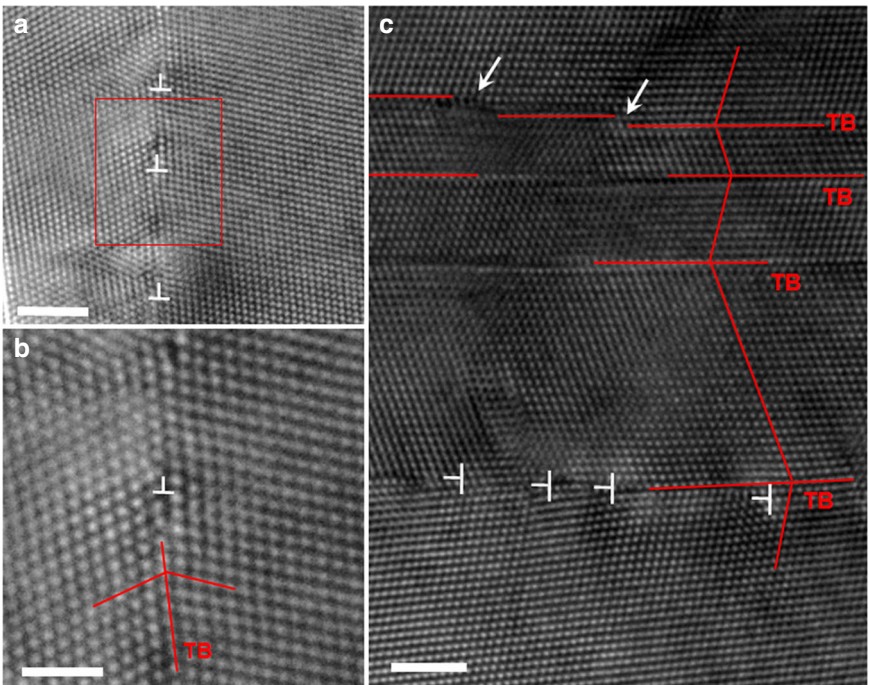

**Fig. 2 The direct observation of switchover from partial dislocation in thinner twins to full dislocation in thick twins. a** A typical HRTEM image showing several full dislocations that intersect with the TBs in relatively thick twins/matrixes (marked with T). **b** Enlarged HRTEM image corresponding to the red framed region of **a**. **c** In a grain that contains both relatively thin and thick twins/matrixes, several full dislocations glide on the planes that intersect with the TBs and are blocked by the TBs, and several steps resulting from partial dislocations are directly detected in relatively thin twins/matrixes. The scale bars are 2 nm, 1 nm and 2 nm for **a**, **b** and **c**, respectively.

with the TBs. Since the two full dislocations are near the regular GB and no dislocation debris was detected on the TBs, the full dislocations must not be emitted from TBs or TB-GB intersections but only emitted from regular GBs.

Figure 2a shows a typical HRTEM image showing several full dislocations that were blocked by the TBs in relatively thick twins/matrixes. Figure 2b shows an enlarged HRTEM image that corresponds to the red framed region of Fig. 2a, which reveals the full dislocations blocked by the TB more clearly. As noted by T, an extra {111} plane was inserted on the TBs, indicating that full dislocation was blocked by the TB. Figure 2c shows a typical HRTEM image of a grain that contains both thin and thick twins/matrixes. In relatively thick twins/matrixes, we can see several full dislocations on the planes that intersect with the TBs and are blocked by the TBs, as marked by T These full dislocations can sometimes react with TBs and in turn lead to new dislocation configurations, such as a Frank dislocation with a Burgers vector of 1/3[111], which segregates with a TB (see other examples in Supplementary Fig. 2–5). In relatively thin twins/matrixes, several steps resulting from partial dislocation that is parallel with TB are directly detected, as marked with arrows.

To investigate the twin thickness effect on the deformation model in twin–structural NC Pt, the large grains that contain relatively thin twin/matrix were also investigated. Figure 3a–c shows three HRTEM images captured 10 s apart with continuous straining of a nanosized grain. Figure 3a shows an HRTEM image taken when the straining was initially loaded on a twin-structured grain, in which a 4 atomic-layer thick twin/matrix with atomically flat TBs is clearly seen. With continued straining, a step in a TB with different positions can be found in Fig. 3b, c (as marked). A detailed analysis of the step shown in Fig. 3b, c indicates that on the left side of the step, there are 4 atomic layers associated with the twin/matrix, while on the right side of the step, the twin/matrix has 5 atomic layers. This structural feature is associated

with a partial dislocation emitted from the right GB-TB intersection that moves towards the left during the straining. The propagation of the partial dislocation creates an additional twined layer behind its movement, resulting in a total of 5 layered twin/matrix sections on the right side of the partial dislocation. Apart from partial dislocations emitting from GB-TB intersections, we also simultaneously observed partial dislocations emitting from regular GBs, which are featured as stacking faults (SF) that were observed in the grains. Figure 3d, e show a pair of HRTEM images captured 50 s apart under tensile loading of another nanograin that contains a 6 atomic-layer twin/matrix. The plastic deformation process causes a 6 atomic-layered twin/matrix to grow into an 8 atomic-layered twin/matrix. Similar to the case shown in Fig. 3a–c, TB migration is caused by the emission of a few partial dislocations from the GB-TB intersections, gliding through the grain along their TB. Interestingly, SFs (as arrowed) are also seen in both HRTEM images, suggesting that there are also partial dislocations emitted from the regular GBs, which is based on the fact that unstrained nanosized grains should contain no dislocations. These partial dislocations emitted from conventional GBs within relatively thin twins/matrixes can be frequently observed (see other examples in Supplementary Figs. 2–5).

Based on extensive in situ and ex situ HRTEM investigations, we found that for grains larger than ~10 nm, the plastic deformation is mediated almost by full dislocations in the twins/matrixes with a TT > ~ 6 nm, and most of them move on the glide planes that intersect with the TBs, thus blocking or reacting with TBs. In relatively thin twins/matrixes, partial dislocations gliding on the planes parallel with the TBs were commonly observed. On this basis, there indeed exists a TT effect on the plasticity mechanisms, i.e., full dislocation gliding and intersecting with TBs in relatively thick twins/matrixes, while partial dislocation nucleation and gliding runs parallel with TBs

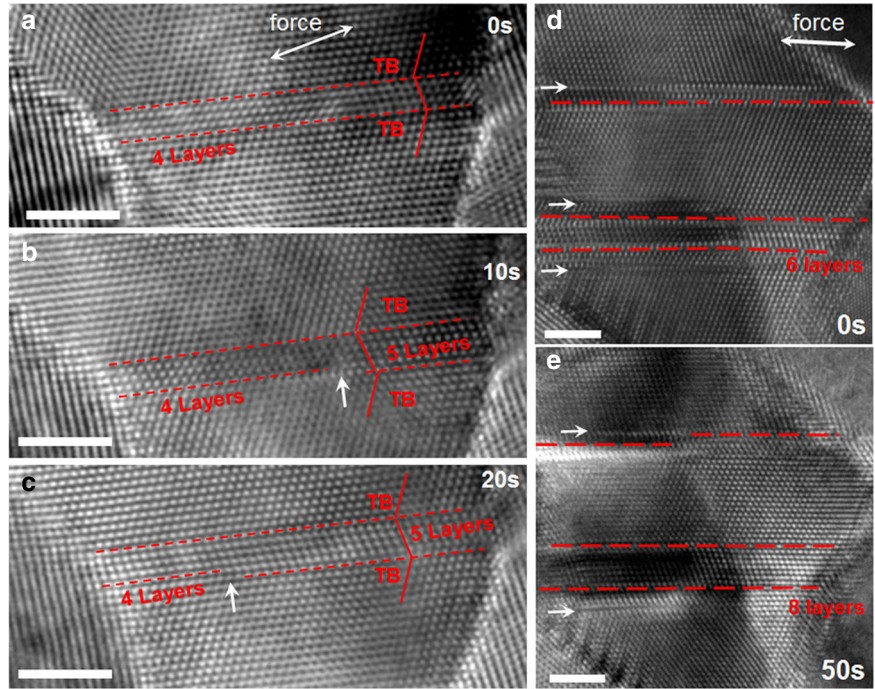

**Fig. 3 In situ observation of partial dislocation emissions both from GB-TB intersection and conventional GBs in a relatively thin twin/matrix. a–c** Partial dislocations are emitted from the GB-TB intersection and glide on the TB, leading to TB migration and formation of a step on the TB. **d** The initial twin/matrix with 6 atomic layers in a grain. **e** An 8 atomic layered twin/matrix was formed. The arrowed SFs are caused by partial dislocation emission from the GBs. The scale bars are for 2 nm.

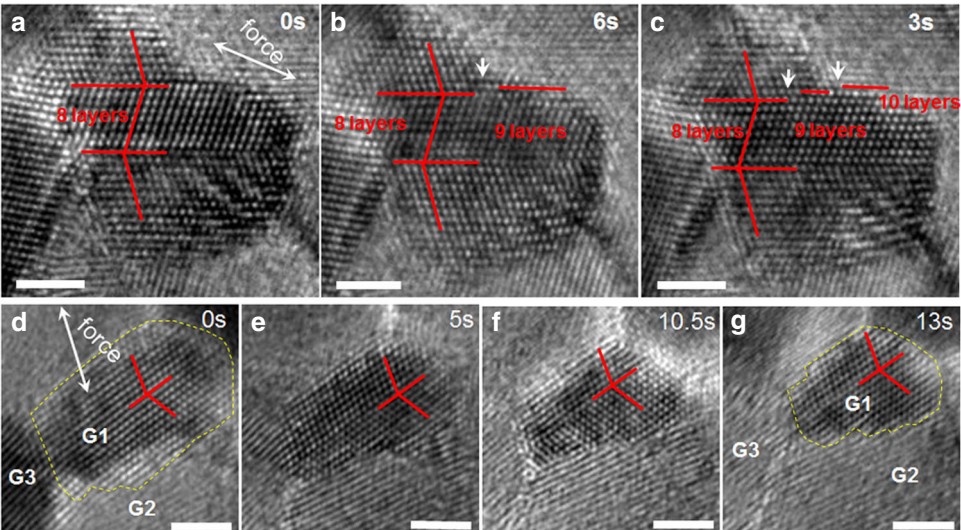

**Fig. 4 Direct observation of switchover from partial dislocation to GB mediated plasticity. a** HRTEM image taken when the loading is initially on the twinned grain, an 8-atomic-layer twin/matrix with atomically flat TBs can be clearly seen. **b** With continued loading, a step resulted from partial dislocation in a TB (as marked). **c** With further straining, two steps resulting from partial dislocation in a TB with different positions can be seen. **d–g** Four HRTEM images captured during continuous tensile loading. The ~4 × 6 nm sized grain shrinks into an ~3 × 4.5 nm sized grain that results from GB migration. The scale bars are for 2 nm.

in relatively thin twins/matrixes (see more examples in Supplementary Figs. 2–9).

To reveal the grain size effect on the deformation model of the twin-structural NC Pt, twin-structural grains with sizes below ~10 nm were also investigated. Figure 4a–c provides a typical example of an in situ observation of partial dislocation in an ~6.5 × 6.5 nm sized grain. Three HRTEM images were captured 3 s apart during tensile loading. Figure 4a shows the loading initially set on the twin-structural grain, and the atomically flat

TBs can be clearly seen. With continued loading, two steps that resulted from two emitted partial dislocations can be found in Fig. 4b, c (as marked by arrows). For grains with *d* less than ~6 nm, our in situ atomic-scale observations show that plastic deformation is controlled by GB-mediated plasticity in lieu of lattice dislocation. Figure 4d–g shows four HRTEM images typical of in situ observations of grain boundary migration in an ~4.0 × 6.0 nm sized grain. Figure 4d shows the initial loading on the twin-structural grain. With continued loading, the grain

shrinks from ~4 × 6 nm to an ~3 × 4.5 nm sized grain, as shown in Fig. 4d–g. In order to indicate the GB migration more clearly, the boundary of G1 between other grains was highlighted using a dotted yellow line. On comparing the grain size of G1 in Fig. 4d, g, it is clear that grain G1 underwent the GB migration. During straining, no change was observed for the lattice in grain $G_1$, indicating that there was no global rotation of the sample. In addition, from Fig. 4d, $G_3$ exhibited an obvious fringe contrast, whereas the fringe nearly disappeared in Fig. 4g, indicating that $G_3$ underwent slight out-of-plane rotation. We also measured the angle between the lattice/fringe between G1, G2 and G3; there was no obvious change, indicating only slight out-of-plane rotation during the GB migration in this case. Apart from out-of-plane rotation during GB migration (also see the other example in Supplementary Fig. 10), GB migration assisted by in-plane rotation was also observed (see the other example in Supplementary Fig. 11) whereas GB sliding was rarely detected. This observation indicates that for the twin-structured NC, there also exists a transition in plasticity mechanisms, i.e., from intra-grain dislocation to a mode of GB-mediated plasticity for grains with sizes below 6 nm (see other examples in Supplementary Figs. 10–12 and the statistical results in Fig. 5).

To reveal the grain size and TT effect on the plasticity transition, we statistically investigated more than 90 twin-structural grains with grain sizes $d$ varying from 4 to ~30 nm and TTs ranging from 1 to ~15 nm. Approximately one-third were observed under in situ investigations, and the rest were observed ex situ. For the in situ observations, the plastic deformation model in twin-structured grains were directly observed until crack nucleation. For the ex situ observation, we examined the twin-structured grains after the thin films fractured. The dislocation resulted in debris in twin-structured grains and the behaviors can be clearly verified by in situ and ex situ HRTEM techniques (see Methods section). Figure 5 shows the statistical results of plasticity events for different twins/matrixes and grains obtained from in situ and ex situ HRTEM observations. For those grains with $d > 6$ nm, the TT is defined as the value of the twin/matrix that underwent dislocation

activities. For example, for a grain containing both thick and thin twins/matrixes, full dislocation was observed in a thick twin/matrix and partial dislocation detected in another thin twin/matrix (Fig. 2c, Supplementary Fig. 2). Following this, we obtained two dots for a given grain size in Fig. 5, where the TT corresponds to the values of the twin/matrix that underwent dislocation activities, respectively. While most of the grains with $d < 6$ nm usually contain only one twin/matrix, we define TT as the average values of the twin and matrix (see Fig. 4d–g and Supplementary Figs. 10–12). We can classify the twin-structural grains into 3 different categories. 1) For $d > ~10$ nm, there exists a transition in deformation mechanisms that occurs at a critical TT; at this point, the plastic deformation from the nucleation and gliding of full dislocation intersecting with the TBs switch to partial dislocations that are parallel with TBs. Interestingly, the critical TT value is different for different grain sizes: the smaller the grain size is, the smaller the critical TT. 2) For $d = 6–9$ nm, partial dislocation nucleation and glide on the planes parallel with the TBs dominate; there is no obvious transition from full dislocation intersecting with TBs to partial dislocation parallel with TBs. 3) For $d < ~6$ nm, the plastic deformation is transferred from lattice dislocation to GB-mediated plasticity, i.e., GB migration. The results indicate that for large-size twin-structural grains (above ~10 nm), the deformation model affects both the grain size and TT, while in small-size twin-structural grains (below ~10 nm), the deformation model only affects the grain size. This is different from the twin-free grains, in which the strengthening and softening mechanisms are only affected by TT.

## Discussion

According to previous experiments[1–5,32–34] and MD predictions[16,18–25,35–37], dislocations gliding and intersecting with TBs lead to strengthening, while dislocations gliding parallel to the TBs result in softening. The statistical HRTEM data show an obvious double size effect on the strengthening and softening deformation model in twin-structured NC metals. First, the grain size controls the strengthening and softening deformation model.

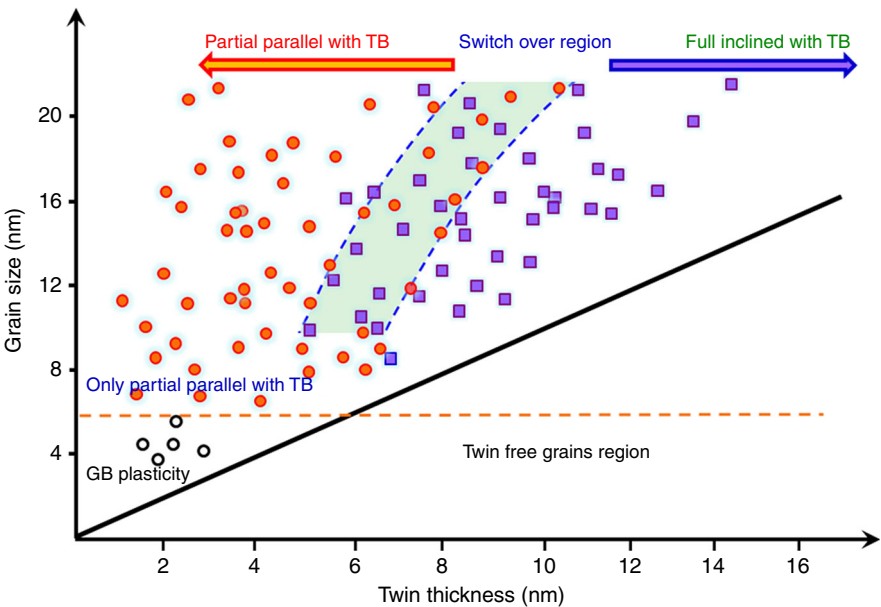

**Fig. 5 Statistical results of the plasticity for twin-structure NC and twin free NC metals.** For $d > ~10$ nm, there is a transition from full dislocation intersecting with the TB switch to partial dislocations parallel with TBs, and the critical TT for the deformation transition is grain-size dependent. For $d = 6–9$ nm, partial dislocation parallel with the TBs dominates. For $d < ~6$ nm, there is a transition from dislocation-controlled plasticity switching to GB-mediated plasticity. The black dots and red squares correspond to statistical results of plasticity events in twin-free NC metals.

As $d$ decreases, there is a transition from dislocation-controlled plasticity switching to GB plasticity-mediated softening that occurs at a grain size of ~6 nm. This transition is similar to twin-free NC metals with grain sizes smaller than ~15 nm[1–5,38,39]. Second, for a given grain size ($d > $~10 nm), the TT can significantly affect the deformation model. For relatively thick twins/matrixes, the plastic deformation is controlled by full dislocation nucleation and intersecting with the TBs, leading to strengthening, while in relatively thin twins/matrixes, partial dislocation nucleation and gliding parallel to TBs lead to softening. More importantly, the value of the critical TT for the transition is different for different grain sizes: the smaller the grain size is, the smaller the critical TT. According to a previous theoretical prediction[16], for a given grain size, the junction of the yielding stress versus TT curve and the Hall-Petch relation indicates the critical TT for the deformation model transition. According to this prediction, the calculated critical TT is ~6.7 nm for the grain with a size of 10 nm, while it is ~8.5 nm for the grain with a size of 20 nm (see Supplementary Note 1 for details)—this is consistent with our statistical results shown in Fig. 5. This indicates that the coupling of grain size and TT's effects on the plastic deformation model that were reported in our experiment and previous theory are universal in twin-structured FCC metals.

Our statistical results show several outcomes that are different from previous MD simulations. First, in previous MD simulations, the deformation model switches involve only partial dislocation, and full dislocation is rarely generated[16,18–25,35–37]. Our statistical experimental results show that the plastic deformation was governed by full dislocations when above the critical TT for a given grain size. The transition from full to partial dislocation in small-sized grains and relatively thin twins/matrixes due to the size effect and the SF energy of Pt (0.27 to 0.373 J/m$^2$) is much higher than those of common FCC metals[40–45]. According to a previous prediction[40], there is a critical $d$ above which full dislocation activities would be preferred over partial dislocation activities, and this critical $d$ can be significantly affected by the SF energy of metals. According to this theory, the calculated critical $d$ is about 33–88 and 7–9.5 nm for Cu and Pt, respectively (see Supplementary Note 2 for details). This is consistent with the statistical result shown in Fig. 5. On this basis, we anticipate that for those metals with high SF energy (such as Pd, Ir), the plastic deformation in large-sized grains and relatively thick twins/matrixes will be governed by full dislocations. For metals with low SF energy, such as Cu, Ag, and Au, partial dislocation can be frequently observed. Second, the TT effect on the strengthening and softening deformation model in our results can only be observed in large-sized grains ($d > $~10 nm); for the grains with $d$ ranging from 9–6 nm, only partial dislocation parallel with TBs was observed, while for grains with $d < $~6 nm, the deformation was controlled by GB plasticity, and no intra-grain dislocation activities were observed. Previous studies did not have a sufficiently small grain size and therefore missed this transition[16,18–25,35–37]. Thus, our in situ atomic-scale results, together with previous atomistic modelling and simulations, advances our physical understanding of the mechanical behaviours of twin-structural NC metals.

Here, the GB migration assisted by GB atomic diffusion and grain rotation is similar to the previous MD simulation in both thin films and 3D NC samples[46,47], where both grain rotation and GB migration were observed. The observed deformation model for grain size below ~6 nm indicated a previously reported switch from dislocation-dominated plasticity to GB-mediated plasticity (inverse Hall-Petch deformation model) in NC metals, which is also valid for twin-structure metals. According to the well-known scaling law of Coble creep[46–48], the relationship between the creep strain rate $\dot{\varepsilon}$ and the grain size $d$ is as follows:

$$\underline{\varepsilon} = k\tau \frac{\Omega \delta D_g}{KTd^3} \tag{1}$$

where $\tau$ is the stress, $\Omega$ is the atomic volume, $\delta$ is the grain boundary width, $D_g$ is the GB diffusivity, $K$ is the Boltzmann constant, and $T$ is the absolute temperature; $k$ is a constant related to grain geometry. According to this equation, the strain rate due to GB diffusion would be enhanced by 3 orders of magnitude as the grain size is reduced by 1 order of magnitude. Thus, the very small grain sizes and high local stress in our twin-structured NC Pt could greatly facilitate GB diffusion-assented GB migration and grain rotation.

Our statistical results demonstrate that 43.4% of grains have dislocations emitted from conventional GBs (as shown in Fig. 1, Fig. 2a, b, Supplementary Figs. 2–5, Supplementary Fig. 6a, Supplementary Fig. 7b, Supplementary Fig. 7c), 34.2% of grains have dislocations emitted from GB-TB intersections (Figs. 3c, 4c, Supplementary Fig. 6b, Supplementary Fig. 7a, Supplementary Fig. 8, Supplementary Fig. 9), and 22.4% of grains have dislocations emitted from both GB-TB intersections and conventional GBs (Figs. 2c, 3d–e). This indicates that conventional GBs are also essential dislocation sources in twin-structural nanograins, which is consistent with MD simulations[16]. To understand why conventional GBs are an important dislocation source, quantitative lattice strain analysis was performed using lattice distortion analysis[49] on the HRTEM images of twin-structural nanograins during the leading process. Figure 6a, b show HRTEM images taken from 2 different twin-structural nanograins captured during the loading process. Figure 6c, d show their corresponding strain maps, and the colour variation corresponds to the different strain values, as indicated in the figures. From the strain maps, strain concentration was observed both near the TB-GB intersection and at the conventional GB. The corresponding quantitative strains, extracted from the coloured strain maps of Fig. 6c, d by using intensity line scanning from top to bottom along the GB (as red line noted), are shown in Fig. 6e, f, respectively. The zero points were denoted by Ref. As noted with arrows, strain that concentrated at both conventional GBs and GB–TB intersections can be observed, and many conventional GBs had an even higher strain concentration than the GB–TB intersections. Considering that the strain was measured before the dislocation emission, the local stress values can be roughly estimated through Young's modulus and multiplied by the elastic strain. From the strain map (Fig. 6e, f), most of the region near GB sustained ~3% strain, which is in the elastic strain region of nanostructured Pt[50]. By assuming Young's modulus of Pt as a constant of 168 GPa, a majority of the region near GB sustained stress of ~5.04 GPa. These high-strain sites are prone to generating dislocations. During straining, the limited GB–TB intersections cannot provide sufficient dislocation nucleation sites; thus, the high-strain sites in conventional GBs become essential sources for generating dislocations[26,32–36] to accommodate large plastic deformations.

It should be noted that beside the grain size and TT can affect the deformation mechanism, the loading orientation can also significantly impact the deformation mechanism of twin-structure materials. Previous studies on Cu revealed that by changing the loading orientation with respect to the TBs, the deformation mechanism switches among three dislocation modes: a dislocation glide between the twins/matrixes, dislocation transfer across TBs, and partial dislocation resulting in TB migration[8]. For this study, we have not considered the effect of the loading direction on the deformation mechanisms as the twin-structural grains in our sample were random distributed without obvious preferred orientation (as shown in Supplementary Fig. 1d). During deformation, the angle between the TBs and

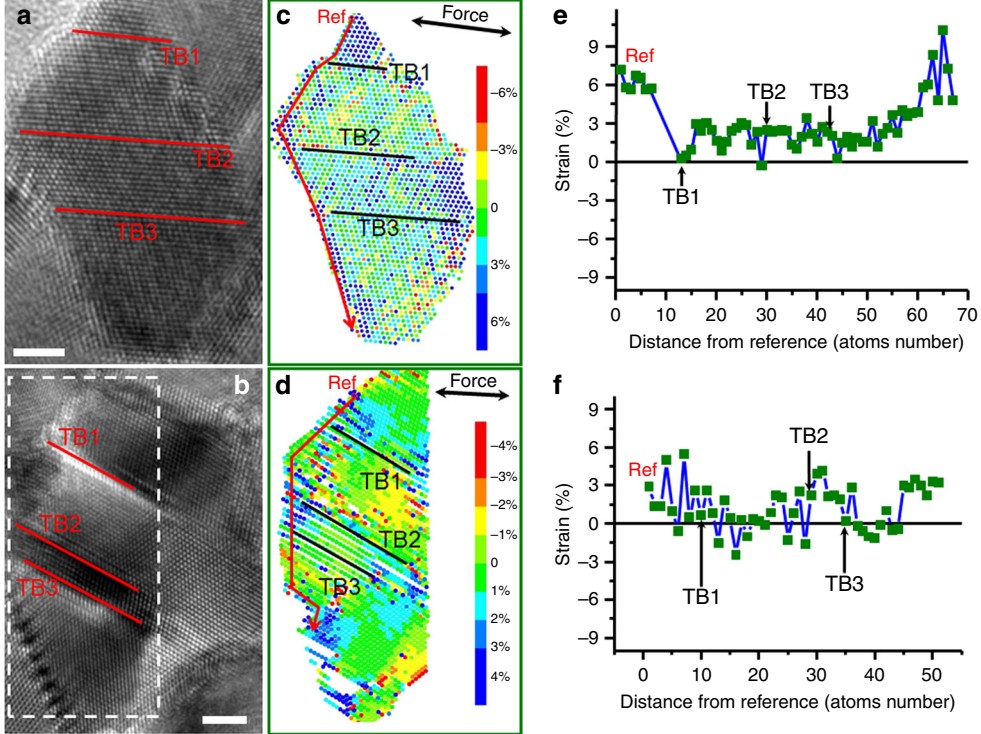

**Fig. 6 Typical strain mapping of twin-structural grains. a, b** Two HRTEM images of the twin-structured grains. **c** The strain mapping corresponds to **a**. **d** The strain mapping that corresponds to the framed region of **b**. As seen from the figures, the strain concentration can be observed both at the conventional GB and GB-TB intersections. **e** Quantitative strains extracted along the red line region of (**c**) by using intensity line scanning from top to bottom along the GB; the zero point is denoted by "Ref". **f** Quantitative strains extracted along the red line region of (**d**) using intensity line scanning from top to bottom along the GB; the zero point is denoted by "Ref". The scale bars are for 2 nm.

the loading direction is varied for different twin-structural grains, which poses a challenge in terms of investigating the effect of loading orientation on the deformation mechanism. The observed grain size and TT effect on the plasticity of NC Pt were similar to that in previous studies on Cu[8,16]. Based on this, we believe that our observed mechanism should be valid for those high-SF energy metals that consist of random distributed twin-structural grains.

In summary, atomic-scale and quickly resolved dynamic deformations in twin-structural Pt nanograins with various grain sizes and TTs have been investigated. The main discoveries are as follows: (1) The experimental atomic-scale plastic deformation provides evidence of twin-structural NC metals that have rarely been evaluated experimentally. (2) There is evidence of a double strengthening and softening deformation model in twin-structural NC metals. (3) As the grain size falls below ~6 nm, the TT effect on the strengthening and softening deformation model is halted and switches to a GB-mediated plasticity. (4) GBs are an important dislocation source for twin-structural NC metals, which can provide enough dislocation nucleation sites to accommodate the large strain. Our findings, together with previous simulations, advance our understanding of the deformation mechanism of twin-structural NC metals, which is useful for designing high-strength yet ductile materials using GBs and TBs.

## Methods

**In situ atomic-scale TEM tensile device**. The homemade TEM tensile stage consists of two thermally actuated bimetallic strips[27–30] fixed in opposing positions on a TEM Cu-ring using epoxy resin. Each bimetallic strip is made of two layers of different materials with a large difference in their thermal expansion coefficients to achieve a significant deflection at a relatively low operating temperature. During the experiment, the bimetallic strips with attached thin films were loaded on a conventional TEM heat stage, the temperature controller accurately increased the temperature (below 80 °C) of the double-tilt TEM heat stage, and the bimetallic extensor then exerted tensile force on the thin films. With this homemade device,

the thin film specimens can be slowly and gently deformed, and the double-tilt capability of the tensile stage can remain. Consequently, the strained grains can be oriented conveniently for high-resolution TEM (HRTEM) observations, and the atomic-scaled deformation process can be recorded during tensile loading. The real-time evolution of the film was captured in situ along with deformation using a JEOL-2010 TEM (operated at 200 kV).

**Transferring thin film to the tensile devices**. The ~15 nm thick Pt thin film was deposited on a Si (001) substrate at 300 °C by magnetron sputtering, and then the Si substrate was etched away using a KOH solution (5 mol/L). Under an optical microscope, the Pt thin film was attached to the bimetallic extension actuator using epoxy resin. Thus, tensile deformation can be ensured during the loading process.

**Dislocation source determination**. HRTEM can directly verify the dislocation sources from the cores of dislocations and their associated SFs/debris in twin-structured grains. When the partial dislocation nucleated from the GB-TB intersection and glided on the plane parallel with the TB, the resulting TB migration or stepped TBs can be detected in grains (as shown in Fig. 3a–c). For the partial dislocations emitted from conventional GBs, whether gliding on the parallel planes or intersecting the TBs, the SF resulting from partial dislocations can be observed, as seen in Fig. 3d, e. For those grains with partial dislocations both emitted from the GB-TB intersection and conventional GBs, both TB migration and SF can be observed. For full dislocation behaviours, the (111) plane inserts an extra plane. All of these dislocation behaviours can be clearly detected by the HRTEM techniques shown in Figs. 1, 2.

## Data availability
The data that support the findings of this study are available from the corresponding author upon reasonable request.

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

## Acknowledgements

This work was supported by the Basic Science Center Program for Multiphase Evolution in Hypergravity of the National Natural Science Foundation of China (No. 51988101), The Beijing Outstanding Young Scientists Projects (BJJWZYJH01201910005018), the NSFC (11722429, 51771104, 91860202), Beijing Natural Science Foundation (Z180014), 111 project (DB18015) and the Fok Ying-Tong Education Foundation of China (151006), and the Australian Research Council (DP190102243).

## Author contributions

L.H.W. and K.D. contributed equally to this work. X.D.H. and Z.Z. designed the project and guided the research. L.H.W. conducted the in situ TEM experiments. J.T. synthesised the thin films. K. D. performed the strain mapping and related discussion. L.H.W., K.D., C.P.Y., J.T., L.B.F., Y.Z.G. and X.D.H. wrote the initial draft. X.D.H. and Z.Z. finalised the paper. All authors contributed to extensive discussions of the results.

## Competing interests

The authors declare no competing interests.
