## [Peer Review File · Nature Communications]

Reviewers' comments:

Reviewer #1 (Remarks to the Author):

With HRTEM observations, the authors explored the dislocation activities in twinned nanocrystalline Pt, with grain size ranges from 4-30nm and twin thickness ranges from 1-15nm.

Although the dislocation mode in fcc twinned metals have been studied extensively and understood quite clearly. The author's work provided strong experimental evidence for the previous theory and simulations. The author successfully found the experimental evidence of the plastic transition from dislocation TB-GB intersection, twinning partials and GB mediated deformation depending on the twin spacing, as well as the grain size.

The results are interesting and provide important experimental data for the studies of the deformation mechanism in nanotwinned metals. The manuscript is publishable on Nature Communications with some revision (see comments).

Some comments need to be addressed:

1. The previous study "You, Z., Li, X., Gui, L., Lu, Q., Zhu, T., Gao, H., & Lu, L. (2013). Plastic anisotropy and associated deformation mechanisms in nanotwinned metals. *Acta Materialia*, 61(1), 217-227."

has shown that by changing the loading orientation with respect to the twin planes, the dominant deformation mechanism can be effectively switched among three dislocation modes, namely dislocation glide in between the twins, dislocation transfer across twin boundaries, and dislocation-mediated boundary migration.

In this study, have you considered the effect of loading direction with respect to the twin direction?

2. For the ultra-small grains (< 6nm), usually how many twins in the grain? Only one? How did you define the twin spacing of this kind of grains in Fig.5?

3. How are the points plotted in Fig.5? Are they based on the full and partials dislocations events observed during the tensile deformation? All the same strain conditions from different samples or from different TEM images at different strains? What happens if the grains contain both thick twins and thin twins?

4. There is some misunderstanding of [16] in the manuscript in line 202. The findings here are not in contrast with the ideas from [16], GBs in nanocrystalline are always one kind of sources for dislocation nucleation.

TB-GB intersections can provide additional sources for twinning partial nucleation, especially when twin spacing is small (a soft mode with lower flow stress).

Reviewer #2 (Remarks to the Author):

In this paper, the authors used the homemade TEM to reveal the underlying atomistic deformation mechanisms in nanocrystalline Pt with nanoscale twins and further summarized the dual effects of grain size and twin thickness on plastic deformation. Especially, the authors, for the first time, showed

the atomic-scale deformation mechanisms in nanocrystalline Pt with nanoscale twins and very small grains, which was not observed in previous studies about nanotwinned metals. Some results provide the mechanistic insights into plastic deformation of nanocrystalline metals with nanoscale twins, and would have significant impact on fabrication and design of nanostructured metals with excellent mechanical properties. Overall, the manuscript is well-organized and well-written. Therefore, the reviewer recommended for publication of this paper. But the authors have to address the following points to further improve the current manuscript.

(1) For fcc metals, the competition between full and partial dislocation activities is related to the stacking fault energy (SFE). Pt has a SFE of about 320 mJ/m², which is higher than those of common fcc metals (such as Au, Ag, Cu, Al, Ni etc). Therefore, the full dislocation activities in Pt are more energetically favorable than the partial dislocation activities. The authors should add the statement about the influence of SFE on full/partial dislocation competition in Pt, and further provides the explanations of why the partial dislocation activities occur in some cases.

(2) Figure 4d-g shows the TEM images of evolution of a tiny grain during deformation, suggesting the grain boundary (GB) mediated mechanisms. It is suggested for the authors to add some statements to indicate the details of GB-mediated mechanisms according to the analysis on grain shape and lattice change based on TEM observations. In the caption of Fig. 4, the authors mentioned that the evolution of tiny grain is related to GB migration. Does it involve GB diffusion and sliding, or grain rotation? What is the driving force for GB migration?

(3) Figure 5 shows the statistical results of TEM observations, and indicates that for a give grain size, there exists a critical twin thickness for the transition in deformation mechanism from full dislocation slip inclined to twin boundaries (TBs) to partial dislocation slip on TBs (i.e. de-twinning). Such phenomenon was observed in nanotwinned Cu with relatively low SFE. TEM observations in the current study suggests that this phenomenon also occurs in Pt with high SFE. It is suggested for the authors to add the relevant discussions to address the coupling of grain-size and twin-thickness effects on plastic deformation of nanotwinned metals and the universality of such phenomenon in fcc metals. Moreover, the quantitative analysis in the current manuscript is relatively lacking. If possible, the authors are suggested to add a theoretical prediction about the critical twin thickness for deformation mechanism transition based on some previous theoretical modelling.

(4) The authors used the strain mapping in Fig. 6 to indicate that the dislocations nucleate from the GB-TB intersections or some sites with strain concentration in GBs. But the dislocations usually nucleate from the stress concentration sites rather than strain concentration sites, although there is a certain stress-strain relationship in the sites for dislocation nucleation. If possible, the authors are suggested to provide a rough estimation of local stress level for dislocation nucleation by making detailed analysis on lattice change from high-resolution TEM observations and the stress-strain relationship.

(5) In the caption of Fig. 3, there are some typo errors, “(a)” and “(b)” in line 4 should be “(d)” and “(e)”, respectively.

Reviewer #3 (Remarks to the Author):

The authors conducted comprehensive In situ observations to clarify the atomic-scale deformation mechanisms of twin-structural nanocrystalline Pt thin films with grain size d in the range of 4-30 nm. The authors reported that (1) at grain size above 10 nm, the deformation mechanism will transform from full dislocation-controlled hardening model to partial dislocation-controlled softening model at below critical twin thickness, and (2) at grain size between 6 and 10 nm, only partial dislocation-

controlled softening model are observed. Finally, (3) at grain size below 6 nm, grain boundary mediated plasticity replaced dislocation-controlled plasticity. The findings, especially at grain size below 10 nm, in this work have great significance, and can enrich our knowledge about the effect of grain size on atomic-scale deformation mechanisms in nanocrystalline and nanotwinned materials. However, there are some concerns before the manuscript can be accepted for publication at Nature Communications:

1. The authors reported double size effect on the strengthening and softening deformation model in nanotwinned Pt. At grain size below 6 nm, the plastic deformation mechanisms is independent of twin thickness. This finding is interesting, yet it is hard to find the explanations in the manuscript. Some discussions about this result should be added, although this may have been comprehensively discussed in the authors' previous works. As compared to previously reported Hall-Petch breakdown in NC metals with extremely small grain sizes, a switch from dislocation dominated plasticity to grain boundary mediated rotation or sliding have been usually proposed. In this work, the authors reported grain boundary migration as the dominating mechanisms at grain size below 6 nm, which is surprising and deserves more attention. Was grain boundary sliding or rotation observed in any of the small grains?
2. The authors also reported that the critical TT where the transition occurs depends on the grain size. In Ref. [16], a model has been proposed to predict the softening and strengthening in nanotwinned and nanocrystalline Cu. Can this model also predict the critical TT at each grain size in Pt?
3. In the discussion section, the authors reported that 43.4% of grains have dislocations emitted from conventional GBs, 34.2% of grains have dislocations emitted from GB-TB intersections, and 22.4% of grains have dislocations emitted from both GB-TB intersections and conventional GBs, and the given evidence is Fig. S8. However, the Fig. S8 just shows the step on the TB caused by partial dislocations emitted from GB-TB intersections, no dislocation nucleation from conventional GB is labeled. Furthermore, while the authors reported that conventional GBs are essential dislocation sources as evidenced from Fig. 6, the statistical results presented by the authors show that the dislocation did not emit from conventional GBs in a considerable number of grains. What is the reason for these results?
4. In the caption of Figure 3, the label (a) and (b) should be (d) and (e).
5. In Fig. 5, each grain size and TT showed exactly one deformation mechanism as represented by a single dot or square. Is it true that for all the grains that have been explored by the authors, only one deformation mechanism was observed for each TT? Is there a case that both full and partial dislocations were nucleated?
6. The authors proposed that the different stacking fault energy in Pt and Cu may be the reason that full dislocations appeared in Pt but not in Cu as in previous findings. The authors need to be more specific on how the SFE may influence the nucleation of full vs. partial dislocations.
7. The use of "thin" and "thick" twin boundaries seem to be arbitrary in the manuscript. For example, the twin thickness shown in Fig. 1 is about 2~3 nm, yet the authors referred to them as "thick".

Point-to-point Response to Reviewer Comments

We thank the reviewers for the in-depth review. The paper has been improved substantially, because of their constructive suggestions. Below we reply to each point raised, first repeating the review comments (in italics) and then following with our response.

Reviewer #1:

General comments: With HRTEM observations, the authors explored the dislocation activities in twinned nanocrystalline Pt, with grain size ranges from 4-30nm and twin thickness ranges from 1-15nm. Although the dislocation mode in fcc twinned metals have been studied extensively and understood quite clearly. The author's work provided strong experimental evidence for the previous theory and simulations. The author successfully found the experimental evidence of the plastic transition from dislocation TB-GB intersection, twinning partials and GB mediated deformation depending on the twin spacing, as well as the grain size. The results are interesting and provide important experimental data for the studies of the deformation mechanism in nanotwinned metals.

Response: We are grateful for the referee's immensely positive comments.

Comment 1: The previous study "You, Z., Li, X., Gui, L., Lu, Q., Zhu, T., Gao, H., & Lu, L. (2013). Plastic anisotropy and associated deformation mechanisms in nanotwinned metals. Acta Materialia, 61(1), 217-227." has shown that by changing the loading orientation with respect to the twin planes, the dominant deformation mechanism can be effectively switched among three dislocation modes, namely dislocation glide in between the twins, dislocation transfer across twin boundaries, and dislocation-mediated boundary migration. In this study, have you considered the effect of loading direction with respect to the twin direction?

Response: We would like to thank the referee for their comments. In fact, before submitting this paper, we carefully read this interesting paper (Reference 8 in our original manuscript). In the current study, the effect of loading orientation on the deformation mechanisms was not considered for the following reasons: (1) Our sample differs from the one published previously (Reference 8). The twinned grains in our sample were randomly distributed without obvious preferred orientation (most TBs were perpendicular to the deposition plane) as shown in Fig. S1d. During deformation, the angle between the TBs and loading direction varies for different twinned grains. (2) Although we know of the global loading direction, the local force is extremely complicated, which makes it a challenge to determine the angle between the force and TBs.

In response to the reference comments, we have labeled the loading direction in our *in situ* TEM images. Moreover, we have added a paragraph to compare our results with this interesting paper. The details of the same are given below:

“It should be noted that beside the grain size and TT can affect the deformation mechanism, the loading orientation can also significantly impact the deformation mechanism of twin-structure materials. Previous studies on Cu revealed that by changing the loading orientation with respect to the TBs, the deformation mechanism switches among three dislocation modes: a dislocation glide between the twins/matrixes, dislocation transfer across TBs, and partial dislocation resulting in TB migration. [8]. For this study, we have not considered the effect of the loading direction on the deformation mechanisms as the twin-structural grains in our sample were random distributed without obvious preferred orientation (as shown in Fig. S1d). During deformation, the angle between the TBs and the loading direction is varied for different twin-structural grains, which poses a challenge in terms of investigating the effect of loading orientation on the deformation mechanism. The observed grain size and TT effect on the plasticity of NC Pt were similar to that in previous studies on Cu [8, 16]. Based on this, we believe that our observed mechanism should be valid for those high-SF energy metals that consist of random distributed twin-structural grains.”

Comment 2: For the ultra-small grains (< 6nm), usually how many twins in the grain? Only one? How did you define the twin spacing of this kind of grains in Fig.5?

Response: We appreciate the referee’s constructive comments. Indeed, most of the grains with $d < 6\text{nm}$ usually contain only one twin/matrix (a few of them contain more than two twins/matrixes). We define the twin thickness as the average values of the twin and matrix (see Fig. 4d–4g and Fig. S10–S12).

In response to the reference comments, we have added several sentences providing a definition of the twin thickness shown in Fig.5. The details are given below:

“For those grains with $d > 6\text{nm}$, TT is defined as the values of the twin/matrix that underwent dislocation activities. For example, for a grain containing both thick and thin twins/matrixes, full dislocation was observed in a thick twin/matrix and partial dislocation was detected in another thin twin/matrix (Fig. 2c, Fig. S2). Following this, we obtained two dots for a given grain size in Figure 5, where the TT corresponded to the values of the twin/matrix that underwent dislocation activities, respectively. While most of the grains with $d < 6\text{nm}$ usually contain only one twin/matrix, we define the TT as the average values of the twin and matrix (see Fig. 4d–4g and Fig. S10–S12).”

Comment 3: How are the points plotted in Fig.5? Are they based on the full and partials dislocations events observed during the tensile deformation? All the same strain conditions from different samples or from different TEM images at different strains? What happens if the grains contain both thick twins and thin twins?

Response: Thanks the referee’s comment. Fig.5 was plotted based on both the *in situ* and *ex situ* HRTEM observation of the plastic events from different TEM images at

different strains. For the *in situ* observations, we pull the thin films until crack nucleation. In the meantime, the plastic deformation model (full/partial or GB plasticity) in twin-structured grains can be directly observed during the pulling process. For each *in situ* observation, we need to conduct a new experiment (the samples used for experiments are synthesized at the same condition; refer to the Methods section), while for the *ex situ* observation, we examined the twin-structured grains after the thin films fractured. For the partial dislocation nucleated from the GB–TB intersection and glided on the plane parallel with the TB, the steps on TBs can be detected (as shown in Fig. 3a–3c). As the partial dislocations are emitted from conventional GBs, whether gliding on the parallel planes or intersecting with the TBs, the stacking faults (SF) resulting from partial dislocations can be observed, as seen in Fig. 3d, 3e. For those grains with partial dislocations emitted both from the GB–TB intersection and conventional GBs, both stepped TBs and SF can be observed. For full dislocation behaviors, we can see that there is an extra plane insert on the (111) plane. Thus, the dislocation behaviors can be clearly detected by *in situ* and *ex situ* HRTEM techniques. For GB plasticity, such as GB migration, sliding, and grain rotation, we need to verify these factors through *in situ* observation.

For the grains containing both relatively thick and thin twins/matrixes, full dislocation activities were usually detected in a relatively thick twin/matrix, and most of these were on the planes intersecting with the TBs, while partial dislocation parallel to TBs were usually detected in relatively thin twins (see Fig. 2c and other examples in Figs. S2). Additionally, only full dislocation in relatively thick twins/matrixes (no partial in thin twin) or partial dislocation in relatively thin twins/matrixes (no full in thick twin) were also observed.

In response to the reference comments, we have added details regarding how Fig.5 was plotted:

“For the *in situ* observations, the plastic deformation model in twin-structured grains were directly observed until crack nucleation. For the *ex situ* observation, we examined the twin-structured grains after the thin films fractured. The dislocation resulted in debris in twin-structured grains and the behaviors can be clearly verified by *in situ* and *ex situ* HRTEM techniques (see methods). Figure 5 shows the statistical results of plasticity events for different twins/matrixes and grains obtained from *in situ* and *ex situ* HRTEM observations. For those grains with $d > 6\text{nm}$, the TT is defined as the value of the twin/matrix that underwent dislocation activities. For example, for a grain containing both thick and thin twins/matrixes, full dislocation was observed in a thick twin/matrix and partial dislocation detected in another thin twin/matrix (Fig. 2c, Fig. S2). Following this, we obtained two dots for a given grain size in Figure 5, where the TT corresponds to the values of the twin/matrix that underwent dislocation activities, respectively. While most of the grains with $d < 6\text{nm}$ usually contain only one twin/matrix, we define TT as the average values of the twin and matrix (see Fig. 4d–4g and Fig. S10–S12).”

Comment 4: *There is some misunderstanding of [16] in the manuscript in line 202. The findings here are not in contrast with the ideas from [16], GBs in nanocrystalline are always one kind of sources for dislocation nucleation. TB-GB intersections can provide additional sources for twinning partial nucleation, especially when twin spacing is small (a soft mode with lower flow stress).*

Response: We have revised the manuscript accordingly.

“Our statistical results demonstrate that 43.4% of grains have dislocations emitted from conventional GBs (as shown in Fig. 1, Fig. 2a, 2b, Fig. S2–S5, Fig. S6a, Fig. S7b, S7c), 34.2% of grains have dislocations emitted from GB-TB intersections (Fig. 3c, Fig. 4c, Fig. S6b, Fig. S7a, Fig. S8, Fig. S9), and 22.4% of grains have dislocations emitted from both GB-TB intersections and conventional GBs (Fig. 2c, Fig. 3d–3e). This indicates that conventional GBs are also essential dislocation sources in twin-structural nanograins, which is consistent with MD simulations [16].”

Reviewer #2:

General comments: *In this paper, the authors used the homemade TEM to reveal the underlying atomistic deformation mechanisms in nanocrystalline Pt with nanoscale twins and further summarized the dual effects of grain size and twin thickness on plastic deformation. Especially, the authors, for the first time, showed the atomic-scale deformation mechanisms in nanocrystalline Pt with nanoscale twins and very small grains, which was not observed in previous studies about nanotwinned metals. Some results provide the mechanistic insights into plastic deformation of nanocrystalline metals with nanoscale twins, and would have significant impact on fabrication and design of nanostructured metals with excellent mechanical properties. Overall, the manuscript is well-organized and well-written. Therefore, the reviewer recommended for publication of this paper.*

Response: We are grateful to the referee for their positive comments.

Comment 1: *For fcc metals, the competition between full and partial dislocation activities is related to the stacking fault energy (SFE). Pt has a SFE of about 320 mJ/m², which is higher than those of common fcc metals (such as Au, Ag, Cu, Al, Ni etc). Therefore, the full dislocation activities in Pt are more energetically favorable than the partial dislocation activities. The authors should add the statement about the influence of SFE on full/partial dislocation competition in Pt, and further provides the explanations of why the partial dislocation activities occur in some cases.*

Response: Following the referee’s suggestion, we have added several sentences in the revised manuscript and supplementary materials. The details are given below:

“The transition from full to partial dislocation in small-sized grains and relatively thin twins/matrixes due to the size effect and the SF energy of Pt (0.27 to 0.373 J/m²) is much higher than those of common FCC metals [40–45]. According to a previous prediction [40], there is a critical d above which full dislocation activities would be preferred over partial dislocation activities, and this critical d can be significantly affected by the SF energy of metals. According to this theory, the calculated critical d is about 33–88 and 7–9.5 nm for Cu and Pt, respectively (see Supplementary Discussion 2 for details). This is consistent with the statistical result shown in Fig. 5. On this basis, we anticipate that for those metals with high SF energy (such as Pd, Ir), the plastic deformation in large-sized grains and relatively thick twins/matrixes will be governed by full dislocations. For metals with low SF energy, such as Cu, Ag, and Au, partial dislocation can be frequently observed.”

Supplementary Discussion 2

The transition from full to partial dislocation. The transition from full to partial dislocation in small-sized grains and thin twins/matrixes can be understood by comparing the shear stress required to nucleate a perfect dislocation to the one required to initiate partial dislocation. According to a previous prediction, the dislocation nucleation stress can be written as follows [4, 5]:

$$\tau_N = 2\mu b_N/d \quad (\text{S1})$$

and

$$\tau_P = 2\mu b_P/d + \gamma/b_P \quad (\text{S2})$$

Where τ_N and τ_P are the critical shear stresses needed to nucleate a full and partial dislocation, d is the grain size, μ is the shear modulus, γ is the stacking fault energy, and b_N and b_P are the magnitudes of the Burgers vectors of the full and partial dislocation, respectively. According to equations (1) and (2), there is a critical grain size d_c , below which the partial dislocation needs a lower stress nucleation than that in full dislocations:

$$d_c = 2\mu(b_N - b_P)b_P/\gamma \quad (\text{S3})$$

For Pt, taking the SF energy as 0.27 to 0.373 J/m² [6, 7] and the shear modulus as 65.2 GPa [8], the calculated critical size d_c is ~ 7–9.5 nm. For Cu, the calculated critical d is about 33–88 nm when taking SF energy of 0.02 to 0.053 J/m [6, 7] and shear modulus of 54.8 GPa [8] into account. For Ag, the calculated critical d is ~ 122.8 nm when taking SF energy of ~ 0.018 J/m² [6, 7] and shear modulus of 56.7 GPa [8] into account.”

Comment 2: Figure 4d-g shows the TEM images of evolution of a tiny grain during deformation, suggesting the grain boundary (GB) mediated mechanisms. It is suggested for the authors to add some statements to indicate the details of GB-mediated mechanisms according to the analysis on grain shape and lattice change based on TEM observations. In the caption of Fig. 4, the authors mentioned that the evolution of tiny grain is related to GB migration. Does it involve GB

diffusion and sliding, or grain rotation? What is the driving force for GB migration?

Response: Following the referee’s suggestion, we have reorganized Fig. 4 and Fig. S10–S12 and added a detailed analysis of GB-mediated mechanisms. Moreover, we have also discussed the driven force of GB-mediated plasticity. The details are provided below:

“In order to indicate the GB migration more clearly, the boundary of G1 between other grains was highlighted using a dotted yellow line. On comparing the grain size of G1 in Fig. 4d and 4g, it is clear that grain G1 underwent the GB migration. During straining, no change was observed for the lattice in grain G₁, indicating that there was no global rotation of the sample. Additionally, from Fig. 4d, G₃ exhibited an obvious fringe contrast, whereas the fringe nearly disappeared in Fig. 4g, indicating that G₃ underwent slight out-of-plane rotation. We also measured the angle between the lattice/fringe between G1, G2, and G3; there was no obvious change, indicating only slight out-of-plane rotation during the GB migration in this case. Apart from out-of-plane rotation during GB migration (also see the other example in Fig. S10), GB migration assisted by in-plane rotation was also observed (see the other example in Fig. S11) whereas GB sliding was rarely detected.”

“Here, the GB migration assisted by GB atomic diffusion and grain rotation is similar to the previous MD simulation in both thin films and 3D NC samples [46, 47], where both grain rotation and GB migration were observed. The observed deformation model for grain size below ~ 6 nm indicated a previously reported switch from dislocation-dominated plasticity to GB-mediated plasticity (inverse Hall-Petch deformation model) in NC metals, which is also valid for twin-structure metals. According to the well-known scaling law of Coble creep

[46–48], the relationship between the creep strain rate $\dot{\epsilon}$ and the grain size d is as follows:

$$\dot{\epsilon} = k t \frac{W d D_g}{K T d^3} \quad (1)$$

where t is the stress, Ω is the atomic volume, δ is the grain boundary width, D_g is the GB diffusivity, K is the Boltzmann constant, and T is the absolute temperature; k is a constant related to grain geometry. According to this equation, the strain rate due to GB diffusion would be enhanced by 3 orders of magnitude as the grain size is reduced by 1 order of magnitude. Thus, the very small grain sizes and high local stress in our twin-structured NC Pt could greatly facilitate GB diffusion-assisted GB migration and grain rotation.”

Comment 3: Figure 5 shows the statistical results of TEM observations, and indicates that for a give grain size, there exists a critical twin thickness for the transition in deformation mechanism from full dislocation slip inclined to twin boundaries (TBs) to partial dislocation slip on TBs (i.e. de-twinning). Such phenomenon was observed in nanotwinned Cu with relatively low SFE. TEM observations in the current study

suggests that this phenomenon also occurs in Pt with high SFE. It is suggested for the authors to add the relevant discussions to address the coupling of grain-size and twin-thickness effects on plastic deformation of nanotwinned metals and the universality of such phenomenon in fcc metals. Moreover, the quantitative analysis in the current manuscript is relatively lacking. If possible, the authors are suggested to add a theoretical prediction about the critical twin thickness for deformation mechanism transition based on some previous theoretical modelling.

Response: Following the referee's suggestion, we have added a theoretical prediction regarding the critical twin thickness for the deformation mechanism transition based on the previous theoretical modelling. The details are provided below:

“According to a previous theoretical prediction [16], for a given grain size, the junction of the yielding stress versus TT curve and the Hall-Petch relation indicates the critical TT for the deformation model transition. According to this prediction, the calculated critical TT is ~ 6.7 nm for the grain with a size of 10 nm, while it is ~ 8.5 nm for the grain with a size of 20 nm (see Supplementary Discussion 1 for details)—this is consistent with our statistical results shown in Fig. 5. This indicates that the coupling of grain size and TT's effects on the plastic deformation model that were reported in our experiment and previous theory are universal in twin-structured FCC metals.”

Supplementary Discussion 1

The critical TT for a given grain size. According to a previous theoretical prediction [1], the dependence of flow stress on both TT and grain size for twin-structured metals is the following:

$$\tau = \frac{KT}{SV} \ln\left[\frac{\lambda e}{dv_D} \exp\left(\frac{VU}{KT}\right)\right] \quad (\text{S1})$$

Where τ is the flow shear stress, K is the Boltzmann constant, T is the temperature, S is a factor representing local stress concentration in the range of ~ 1.2, V is dislocation activation volume, v_D is the Debye frequency in the range of $1.0 \times 10^{13}/s$, ΔU is the activation energy, λ is TT, e is strain rate, and d is grain size.

The Hall-Petch dependence of flow stress on grain size is as follows:

$$\tau = \tau_0 + \frac{k}{\sqrt{d}} \quad (\text{S2})$$

According to previous experimental studies, the τ_0 is about 0.001G, where G is

Young's modulus of materials. $k = 3266 \text{MPa} \sqrt{\text{nm}}$ and this is nearly the same for

the FCC metals [2].

According to equation (1) and (2), the critical TT for a given grain size is as follows:

$$\frac{k}{\sqrt{d}} + \frac{KT}{SV} \ln\left(\frac{dv_D}{\lambda e}\right) = \frac{VU}{SV} - \tau_0 \quad (\text{S3})$$

In our experiment, T is $\sim 330\text{K}$ and e is $\sim 10^{-3}$. According to previous theory [3], V is $\sim 3b^3$ and ΔU is 1.2eV . According to equation (3), the calculated critical TT is $\sim 6.7\text{ nm}$ for the grain with a size of 10 nm , while it is $\sim 8.5\text{nm}$ for the grain with a size of 20nm .

Comment 4: The authors used the strain mapping in Fig. 6 to indicate that the dislocations nucleate from the GB-TB intersections or some sites with strain concentration in GBs. But the dislocations usually nucleate from the stress concentration sites rather than strain concentration sites, although there is a certain stress-strain relationship in the sites for dislocation nucleation. If possible, the authors are suggested to provide a rough estimation of local stress level for dislocation nucleation by making detailed analysis on lattice change from high-resolution TEM observations and the stress-strain relationship.

Response: Following the referee's advice, we have roughly estimated the local stress. The details are provided below:

“Considering that the strain was measured before the dislocation emission, the local stress values can be roughly estimated through Young's modulus and multiplied by the elastic strain. From the strain map (Fig. 6e and Fig. 6f), most of the region near GB sustained $\sim 3\%$ strain, which is in the elastic strain region of nanostructured Pt [50]. By assuming Young's modulus of Pt as a constant of 168 GPa , a majority of the region near GB sustained stress of $\sim 5.04\text{ GPa}$.”

Comment 5: In the caption of Fig. 3, there are some typo errors, “(a)” and “(b)” in line 4 should be “(d)” and “(e)”, respectively.

Response: We have revised the manuscript accordingly.

Reviewer #3:

General comments: The authors conducted comprehensive In situ observations to clarify the atomic-scale deformation mechanisms of twin-structural nanocrystalline Pt thin films with grain size d in the range of $4\text{-}30\text{ nm}$. The authors reported that (1) at grain size above 10 nm , the deformation mechanism will transform from full dislocation-controlled hardening model to partial dislocation-controlled softening model at below critical twin thickness, and (2) at grain size between 6 and 10 nm , only partial dislocation-controlled softening model are observed. Finally, (3) at grain

size below 6 nm, grain boundary mediated plasticity replaced dislocation-controlled plasticity. The findings, especially at grain size below 10 nm, in this work have great significance, and can enrich our knowledge about the effect of grain size on atomic-scale deformation mechanisms in nanocrystalline and nanotwinned materials.

Response: We are grateful for the referee's very positive comments.

Comment 1: *The authors reported double size effect on the strengthening and softening deformation model in nanotwinned Pt. At grain size below 6 nm, the plastic deformation mechanisms is independent of twin thickness. This finding is interesting, yet it is hard to find the explanations in the manuscript. Some discussions about this result should be added, although this may have been comprehensively discussed in the authors' previous works. As compared to previously reported Hall-Petch breakdown in NC metals with extremely small grain sizes, a switch from dislocation dominated plasticity to grain boundary mediated rotation or sliding have been usually proposed. In this work, the authors reported grain boundary migration as the dominating mechanisms at grain size below 6 nm, which is surprising and deserves more attention. Was grain boundary sliding or rotation observed in any of the small grains?*

Response: We thank the referee for the constructive suggestion. We have reorganized Fig. 4 and Fig. S10–S12 and have added a detailed analysis of GB-mediated mechanisms. Moreover, we have also discussed the driven force of GB-mediated plasticity. The details are given below:

“In order to indicate the GB migration more clearly, the boundary of G1 between other grains was highlighted using a dotted yellow line. On comparing the grain size of G1 in Fig. 4d and 4g, it is clear that grain G1 underwent the GB migration. During straining, no change was observed for the lattice in grain G₁, indicating that there was no global rotation of the sample. Additionally, from Fig. 4d, G₃ exhibited an obvious fringe contrast, whereas the fringe nearly disappeared in Fig. 4g, indicating that G₃ underwent slight out-of-plane rotation. We also measured the angle between the lattice/fringe between G1, G2, and G3; there was no obvious change, indicating only slight out-of-plane rotation during the GB migration in this case. Apart from out-of-plane rotation during GB migration (also see the other example in Fig. S10), GB migration assisted by in-plane rotation was also observed (see the other example in Fig. S11) whereas GB sliding was rarely detected.”

“Here, the GB migration assisted by GB atomic diffusion and grain rotation is similar to the previous MD simulation in both thin films and 3D NC samples [46, 47], where both grain rotation and GB migration were observed. The observed deformation model for grain size below ~ 6 nm indicated a previously reported switch from dislocation-dominated plasticity to GB-mediated plasticity (inverse Hall-Petch deformation model) in NC metals, which is also valid for twin-structure metals. According to the well-known scaling law of Coble creep

[46–48], the relationship between the creep strain rate $\dot{\epsilon}$ and the grain size d is

$$\dot{\epsilon} = k t \frac{W d D}{K T d^3}$$

as follows:

(1)

where t is the stress, Ω is the atomic volume, δ is the grain boundary width, D_g is the GB diffusivity, K is the Boltzmann constant, and T is the absolute temperature; k is a constant related to grain geometry. According to this equation, the strain rate due to GB diffusion would be enhanced by 3 orders of magnitude as the grain size is reduced by 1 order of magnitude. Thus, the very small grain sizes and high local stress in our twin-structured NC Pt could greatly facilitate GB diffusion-assisted GB migration and grain rotation.”

Comment 2: The authors also reported that the critical TT where the transition occurs depends on the grain size. In Ref. [16], a model has been proposed to predict the softening and strengthening in nanotwinned and nanocrystalline Cu. Can this model also predict the critical TT at each grain size in Pt?

Response: Following the referee’s construction suggestion, we have added a theoretical prediction regarding the critical twin thickness for the deformation mechanism transition based on previous theoretical modelling. The details are provided below:

“According to a previous theoretical prediction [16], for a given grain size, the junction of the yielding stress versus TT curve and the Hall-Petch relation indicates the critical TT for the deformation model transition. According to this prediction, the calculated critical TT is ~ 6.7 nm for the grain with a size of 10 nm, while it is ~ 8.5 nm for the grain with a size of 20 nm (see Supplementary Discussion 1 for details)—this is consistent with our statistical results shown in Fig. 5. This indicates that the coupling of grain size and TT’s effects on the plastic deformation model that were reported in our experiment and previous theory are universal in twin-structured FCC metals.”

Supplementary Discussion 1

The critical TT for a given grain size. According to a previous theoretical prediction [1], the dependence of flow stress on both TT and grain size for twin-structured metals is the following:

$$\tau = \frac{KT}{SV} \ln\left[\frac{\lambda e}{d\nu_D} \exp\left(\frac{VU}{KT}\right)\right] \quad (\text{S1})$$

Where τ is the flow shear stress, K is the Boltzmann constant, T is the temperature, S is a factor representing local stress concentration in the range of ~ 1.2, V is dislocation activation volume, ν_D is the Debye frequency in the range of $1.0 \times 10^{13}/s$, ΔU is the activation energy, λ is TT, $\dot{\epsilon}$ is strain rate, and d is grain size.

The Hall-Petch dependence of flow stress on grain size is as follows:

$$\tau = \tau_0 + \frac{k}{\sqrt{d}} \quad (\text{S2})$$

According to previous experimental studies, the τ_0 is about 0.001G, where G is Young's modulus of materials. $k = 3266\text{MPa}\sqrt{\text{nm}}$ and this is nearly the same for the FCC metals [2].

According to equation (1) and (2), the critical TT for a given grain size is as follows:

$$\frac{k}{\sqrt{d}} + \frac{KT}{SV} \ln\left(\frac{dv_D}{\lambda e}\right) = \frac{VU}{SV} - \tau_0 \quad (\text{S3})$$

In our experiment, T is $\sim 330\text{K}$ and e is $\sim 10^{-3}$. According to previous theory [3], V is $\sim 3b^3$ and ΔU is 1.2ev . According to equation (3), the calculated critical TT is $\sim 6.7\text{ nm}$ for the grain with a size of 10 nm , while it is $\sim 8.5\text{nm}$ for the grain with a size of 20nm .

Comment 3: In the discussion section, the authors reported that 43.4% of grains have dislocations emitted from conventional GBs, 34.2% of grains have dislocations emitted from GB-TB intersections, and 22.4% of grains have dislocations emitted from both GB-TB intersections and conventional GBs, and the given evidence is Fig. S8. However, the Fig. S8 just shows the step on the TB caused by partial dislocations emitted from GB-TB intersections, no dislocation nucleation from conventional GB is labeled. Furthermore, while the authors reported that conventional GBs are essential dislocation sources as evidenced from Fig. 6, the statistical results presented by the authors show that the dislocation did not emit from conventional GBs in a considerable number of grains. What is the reason for these results?

Response: We regret the misunderstanding and ambiguity caused by our writing style. HRTEM can verify the dislocation sources from the cores of dislocations and their associated SFs/debris in twin-structured grains. For the partial dislocations emitted from conventional GBs, the SF resulting from dislocations can be observed in the grain, while full dislocation resulting in the (111) plane inserts an extra plane and the dislocation source can be clearly determined according to its gliding plane (as shown in Fig. 1, Fig. 2a, 2b, Fig. S2–S5, Fig. S6a, Fig. S7b, S7c). When the partial dislocation is nucleated from the GB–TB intersection, the resulting TB migration or stepped TBs can be detected in grains (Fig. 3c, Fig. 4c, Fig. S6b, Fig. S7a, Fig. S8, Fig. S9). For those grains with partial dislocations emitted both from the GB–TB intersection and conventional GBs, TB migration and SF can be observed (Fig. 2c, Fig. 3d–3e).

Following the referee's suggestion, we have revised this sentence accordingly:

“Our statistical results demonstrate that 43.4% of grains have dislocations emitted from conventional GBs (as shown in Fig. 1, Fig. 2a, 2b, Fig. S2–S5, Fig. S6a, Fig. S7b, S7c), 34.2% of grains have dislocations emitted from GB-TB intersections (Fig. 3c, Fig. 4c, Fig. S6b, Fig. S7a, Fig. S8, Fig. S9), and 22.4% of grains have dislocations emitted from both GB-TB intersections and conventional GBs (Fig. 2c, Fig. 3d–3e). This indicates that conventional GBs are also essential dislocation sources in twin-structural nanograins, which is consistent with MD simulations [16].”

Comment 4: In the caption of Figure 3, the label (a) and (b) should be (d) and (e).

Response: We have revised the manuscript accordingly.

Comment 5: In Fig. 5, each grain size and TT showed exactly one deformation mechanism as represented by a single dot or square. Is it true that for all the grains that have been explored by the authors, only one deformation mechanism was observed for each TT? Is there a case that both full and partial dislocations were nucleated?

Response: Indeed, we had not observed that both full and partial dislocations were nucleated in the same twin/matrix, although full and partial dislocations nucleated in different twin/matrixes with nearly the same thickness (in the same grain) were observed (Fig. S5). In our experiments, for the grains containing both relatively thick and thin twin/matrixes, full dislocation activities were usually detected in relatively thick twins/matrixes, while partial dislocation parallel with TB were usually detected in relatively thin twins/matrixes (see Fig. 2c and other examples in Figs. S2).

Comment 6: The authors proposed that the different stacking fault energy in Pt and Cu may be the reason that full dislocations appeared in Pt but not in Cu, as in previous findings. The authors need to be more specific on how the SFE may influence the nucleation of full vs. partial dislocations.

Response: Following the referee's constructive suggestion, we have added several sentences in the revised manuscript and supplementary materials. The details are given below:

“The transition from full to partial dislocation in small-sized grains and relatively thin twins/matrixes due to the size effect and the SF energy of Pt (0.27 to 0.373 J/m²) is much higher than those of common FCC metals [40–45]. According to a previous prediction [40], there is a critical d above which full dislocation activities would be preferred over partial dislocation activities, and this critical d can be significantly affected by the SF energy of metals. According to this theory, the calculated critical d is about 33–88 and 7–9.5 nm for Cu and Pt, respectively (see Supplementary Discussion 2 for details). This is consistent with the statistical result shown in Fig. 5. On this basis, we anticipate that for those metals with high SF energy (such as Pd, Ir), the plastic deformation in

large-sized grains and relatively thick twins/matrixes will be governed by full dislocations. For metals with low SF energy, such as Cu, Ag, and Au, partial dislocation can be frequently observed.”

Supplementary Discussion 2

The transition from full to partial dislocation. The transition from full to partial dislocation in small-sized grains and thin twins/matrixes can be understood by comparing the shear stress required to nucleate a perfect dislocation to the one required to initiate partial dislocation. According to a previous prediction, the dislocation nucleation stress can be written as follows [4, 5]:

$$\tau_N = 2\mu b_N/d \quad (S1)$$

and

$$\tau_P = 2\mu b_P/d + \gamma/b_P \quad (S2)$$

Where τ_N and τ_P are the critical shear stresses needed to nucleate a full and partial dislocation, d is the grain size, μ is the shear modulus, γ is the stacking fault energy, and b_N and b_P are the magnitudes of the Burgers vectors of the full and partial dislocation, respectively. According to equations (1) and (2), there is a critical grain size d_c , below which the partial dislocation needs a lower stress nucleation than that in full dislocations:

$$d_c = 2\mu(b_N - b_P)b_P/\gamma \quad (S3)$$

For Pt, taking the SF energy as 0.27 to 0.373 J/m² [6, 7] and the shear modulus as 65.2 GPa [8], the calculated critical size d_c is ~ 7–9.5 nm. For Cu, the calculated critical d is about 33–88 nm when taking SF energy of 0.02 to 0.053 J/m [6, 7] and shear modulus of 54.8 GPa [8] into account. For Ag, the calculated critical d is ~ 122.8 nm when taking SF energy of ~ 0.018 J/m² [6, 7] and shear modulus of 56.7 GPa [8] into account.”

Comment 7: The use of "thin" and "thick" twin boundaries seems to be arbitrary in the manuscript. For example, the twin thickness shown in Fig. 1 is about 2~3 nm, yet the authors referred to them as "thick".

Response: We have revised the manuscript according to the referee’s suggestion. Additionally, we also have revised the “thin” and “thick” twins as the “relatively thin twin/matrix” and “relatively thick twin/matrix,” respectively.

REVIEWERS' COMMENTS:

Reviewer #1 (Remarks to the Author):

The authors have addressed all my comments and revised the manuscript accordingly. The current manuscript is recommended for publication.

Reviewer #2 (Remarks to the Author):

After carefully reading the responses from the authors and the revised manuscript, I felt that the authors have addressed all comments from there reviewers and the manuscript has been indeed improved much. The results from the current study would have significant impact on the fundamental understanding of plastic deformation of nanostructured metals and the design and fabrication of nanostructured metals with excellent mechanical properties. Therefore, the reviewer strongly recommended this paper for publication. But there is only one typo error which needs to be corrected. In the line 89, page 8 of Supplementary Materials, the unit of Delta_U should be “eV” not “ev”.

Reviewer #3 (Remarks to the Author):

The authors have satisfactorily addressed all my previous concerns and I would recommend acceptance of the manuscript at Nature Communications.

Point-to-point Response to Reviewer Comments

We thank the reviewers for the in-depth review. Below we reply to each point raised, first repeating the review comments (in italics) and then following with our response.

Reviewer #1:

General comments: The authors have addressed all my comments and revised the manuscript accordingly. The current manuscript is recommended for publication.

Response: We are grateful for the referee's positive comments.

Reviewer #2:

General comments: After carefully reading the responses from the authors and the revised manuscript, I felt that the authors have addressed all comments from there reviewers and the manuscript has been indeed improved much. The results from the current study would have significant impact on the fundamental understanding of plastic deformation of nanostructured metals and the design and fabrication of nanostructured metals with excellent mechanical properties. Therefore, the reviewer strongly recommended this paper for publication. But there is only one typo error which needs to be corrected.

Response: We are grateful for the referee's positive comments.

Comments 1: In the line 89, page 8 of Supplementary Materials, the unit of Delta_U should be "eV" not "ev".

Response: We have revised Supplementary Materials accordingly.

Reviewer #3:

General comments: The authors have satisfactorily addressed all my previous concerns and I would recommend acceptance of the manuscript at Nature Communications.

Response: We are grateful for the referee's very positive comments.